# Multi-Camera-Based Sorting System for Surface Defects of Apples

**DOI:** 10.3390/s23083968

**Published:** 2023-04-13

**Authors:** Ju-Hwan Lee, Hoang-Trong Vo, Gyeong-Ju Kwon, Hyoung-Gook Kim, Jin-Young Kim

**Affiliations:** 1Department of ICT Convergence System Engineering, Chonnam National University, 77 Yongbong-ro, Buk-gu, Gwangju 61186, Republic of Korea; juhwanlee@jnu.ac.kr (J.-H.L.); hoangtrong2305@gmail.com (H.-T.V.); 2LINUXIT, 53-18, Geumbong-ro 44beon-gil, Gwangsan-gu, Gwangju 62377, Republic of Korea; gyeongju@linuxit.co.kr; 3Department of Electronic Convergence Engineering, Kwangwoon University, 20 Gwangun-ro, Nowon-gu, Seoul 01897, Republic of Korea; hkim@kw.ac.kr

**Keywords:** apple sorting system, multi-camera, CNN classifier, knowledge distillation

## Abstract

In this paper, we addressed the challenges in sorting high-yield apple cultivars that traditionally relied on manual labor or system-based defect detection. Existing single-camera methods failed to uniformly capture the entire surface of apples, potentially leading to misclassification due to defects in unscanned areas. Various methods were proposed where apples were rotated using rollers on a conveyor. However, since the rotation was highly random, it was difficult to scan the apples uniformly for accurate classification. To overcome these limitations, we proposed a multi-camera-based apple sorting system with a rotation mechanism that ensured uniform and accurate surface imaging. The proposed system applied a rotation mechanism to individual apples while simultaneously utilizing three cameras to capture the entire surface of the apples. This method offered the advantage of quickly and uniformly acquiring the entire surface compared to single-camera and random rotation conveyor setups. The images captured by the system were analyzed using a CNN classifier deployed on embedded hardware. To maintain excellent CNN classifier performance while reducing its size and inference time, we employed knowledge distillation techniques. The CNN classifier demonstrated an inference speed of 0.069 s and an accuracy of 93.83% based on 300 apple samples. The integrated system, which included the proposed rotation mechanism and multi-camera setup, took a total of 2.84 s to sort one apple. Our proposed system provided an efficient and precise solution for detecting defects on the entire surface of apples, improving the sorting process with high reliability.

## 1. Introduction

High-yield apple cultivars undergo initial manual labor or system-based defect detection for grading and sorting in preparation for commercial sale. However, manual labor can inevitably lead to misclassification due to human fatigue over prolonged periods. To achieve uniform classification, robots or systems based on cameras or optical sensors and machine learning or algorithm-based technologies [1,2] were developed to perform sorting tasks. Camera-based vision systems observe external apple features, such as deformities, coloration rates, and defect areas, while optical lens-based systems, such as hyper-spectral cameras [3,4], analyze internal apple characteristics that are not observable from the surface by examining the wavelengths of hyperspectral images. These specialized systems are not suitable for direct use in orchards or farms due to their high cost and the need for expert analysis. Consequently, various apple defects detection systems propose camera-based algorithms and hardware to address these challenges in sorting.

One simpler method [5] involves mounting a camera above the apples to capture surface images, which are then analyzed using deep learning. However, due to the structural characteristics of a single-camera system, only unidirectional imaging is possible, analyzing and detecting only one side of the apple without any physical rotation. To address this limitation, systems have been proposed [6,7,8,9] that use conveyor belts and moving mechanisms to rotate the apples as they are transported. Although this allows the camera to evenly scan their surfaces, it may not perfectly scan defects on the apple’s skin if a mechanism for uniformly capturing the entire surface of the apple is not proposed. In fact, apples are not uniformly shaped, and, in the case of defective apples, they may have irregular, elliptical forms. As a result, there is no guarantee that rolling randomly will uniformly scan the entire surface. Furthermore, if minor damages, such as scratches, are located in difficult-to-scan blind spots, the system’s reliability may be compromised.

To address these issues, we propose a sorting system that can uniformly and accurately capture the entire surface of apples using an appropriate rotation mechanism and a multi-camera setup. Our rotation mechanism does not rely on rolling on a conveyor belt, as in previous studies, but, instead, applies rotation to individual objects. To uniformly capture the entire surface of the rotating apple, three cameras are used simultaneously. Using three cameras has the advantage of quickly and uniformly acquiring the entire surface in conjunction with the rotation mechanism, as opposed to using a single camera. The images obtained in this manner are then analyzed using a deep learning model deployed on embedded hardware.

The superiority of deep learning models has already been demonstrated in the field of image classification. High-performing models in image classification are primarily based on CNNs [10,11,12], which have an increased depth, or are more recently designed with transformer-based architectures [13,14], which tend to have a large number of parameters. As the size of models increases, various research efforts have emerged to make them usable on limited hardware resources [15]. Techniques such as pruning [16] and quantization [17,18] aim to reduce the number of parameters and maintain performance through model compression. However, as the complexity of deep learning model architectures increases, methods that directly modify model weights, such as pruning, can prolong model development time from an engineering perspective. Therefore, this study employs knowledge distillation [19], which offers a relatively easy-to-implement approach while ensuring performance. Knowledge distillation exhibits exceptional advantages in terms of “model size reduction”, “inference speed improvement”, and “enhanced generalization performance”.

Our study presents three significant contributions. Firstly, we introduce a multi-camera system that rotates regularly, allowing for the detection of defects on the entire surface of apples. This system captures high-quality images of apples from all angles and provides more precise defect detection than traditional inspection methods. Secondly, we develop a segmentation preprocessing technique that improves the accuracy of the CNN classifier used for defect detection in apples. By segmenting the apples into smaller regions, we can more accurately identify defects and reduce the noise. Finally, we enhance the processing speed and accuracy of the embedded system by using knowledge distillation. By transferring knowledge from a larger model to a smaller one, we can achieve comparable performance with significantly fewer computational resources. In summary, our proposed system offers an efficient and precise solution for detecting defects in apples.

## 2. Related Works

Apple sorting systems commonly employ conveying systems to transfer apples and mount cameras above the conveyor to capture and process images of the passing objects. The conveying mechanisms can vary in their mode of operation, including linear or rotational motion, while the cameras can differ in their quantity, with some systems employing a single camera and others utilizing multiple cameras. Accordingly, current apple sorting systems can be classified into four categories: S-R (single camera with rotation mechanism), S-N/R (single camera with non-rotation mechanism), M-R (multi-camera with rotation mechanism), and M-N/R (multi-camera with non-rotation mechanism). As detailed in the comparison table presented in [9], some sorting systems may not incorporate cameras or may use NIR cameras. However, this study focuses specifically on utilizing RGB images for image analysis and does not address those systems separately. S-N/R and M-N/R systems only scan one side of the apple and classify them as “normal/defective” or based on a grading scale of “high/middle/low”. However, scanning only one side of the apple fails to uniformly assess the entire surface, rendering it unsuitable for scanning the entire surface. Although M-N/R may appear to consider multiple sides, it does not evaluate the bottom side, making it difficult to claim that the entire surface was scanned.

Therefore, employing proper rotation mechanisms during apple transportation, as seen in S-R and M-R systems, enables the scanning of the entire apple surface. However, such a transportation process may cause the apples to rotate too quickly and freely for the system to claim consistent scanning of the entire surface. Various previous studies, as presented in Table 1, utilize different hardware setups and classifiers, but they lack the capability to uniformly scan the apple surface. Therefore, this study utilizes a straightforward hardware approach but incorporates multi-cameras and rotation mechanisms to comprehensively scan the apple using six 2D images.

## 3. Materials and Methods

### 3.1. System Design

The apple sorting system consists of three primary sections: the conveying section, classification section, and holder section. A computer with an Intel^®^ Core™ i5-11400 CPU, 32 GB of DDR4-3200 memory, a 2 TB system storage, and a 4 TB data storage controlled the entire system.

The overall process of the apple sorting system is depicted in Figure 1a–f. Apples are loaded onto the conveyor (Figure 1a) and placed into individual holders for scanning and classification (Figure 1b). The apples detected by the object detection sensor roll towards the holder and settle into place (Figure 1c). The process of apple insertion into the holder is controlled by the object detection sensor, and it is indicated in red circles in Figure 1b,c. Once in the holder, the apple is immediately captured by the cameras, and the obtained images are processed through the CNN classifier deployed on each embedded board for inference. The apple is then rotated once, and the same inference is performed for the captured images on the other side of the apple (Figure 1d). More information about the rotation and camera structure can be found in the description in the Holder section. Finally, the stepping motor moves the apple to the box according to its classification results in the direction of the red arrow (Figure 1e), and the results can be monitored (Figure 1f). To aid in understanding, the workflow of the system is depicted in Figure 2.

Moving to the components of the system, the conveying section consisted of a conveying belt, measuring 2300 mm (W) × 1000 mm (H) × 700 mm (D), divided into an apple supply conveyor and an apple transfer conveyor belt. Apples were transferred to the holder through the fruit supply conveyor belt, and their movement flow was regulated by the central sensor. The central sensor could manage the general fault condition of the system by halting the motor drive immediately to prevent bottlenecking during apple input.

The classification section comprised an industrial camera, a light sensor, and an embedded board (NVIDIA Jetson Nano [30]). The IMX264 CMOS sensing device used an industrial camera (acA2240-35uc, Basler AG, Schleswig-Holstein, Germany) with a 5 MP resolution (2448 × 2048 pixels) and a speed of 35 frames per second at 5.0 MP resolution. The camera was equipped with a lens (FL-CC1618-5MX, Ricoh Co., Ltd., Tokyo, Japan) with a focal length of 16 mm and a maximum aperture ratio of 1:1.8. To obtain uniform and accurate images of apple surfaces with an elliptical shape, we designed a camera structure with three cameras placed at a 120-degree interval (Figure 3a). To minimize light interference, the cameras were placed under illumination.

In the holder section, an apple detected by the object detection sensor rolled into the holder, and three cameras simultaneously captured each of the three sides (Figure 3a). The holder then rotated the apple 180 degrees in the direction of the red arrow using a rotation mechanism based on the principle of rotational friction force generated by rapidly running the motor of the internal roller (Figure 3b). After rotation, there was a two-second waiting period for the apple to stabilize in the holder, which prevented significant motion blur from the camera during rotation and enhanced data quality for classification. Once the apple had stabilized, all three cameras captured its three sides again, resulting in six images of the apple’s surface: three before rotation and three after rotation. The three images before the rotation were immediately classified by the CNN classifier on the embedded boards. The classification results of the three images after the rotation were obtained by the same process. After obtaining a total of six classification results, if even one image detects a surface defect, the external stepping motor of the holder rotated 45 degrees to sort the apple into the box (Figure 3c). Therefore, this is to rotate to the Surface-defects box as indicated by the red arrow on the left, and rotate to the normal box as indicated by the red arrow on the right.

However, it was necessary to verify the reliability of whether the designed camera structure captures the entire surface of the apple. Therefore, a random sample of apples with an average diameter of 150 mm or less was collected to verify whether the surface of the apple was uniformly captured. To quantify the surface exposure rate, the surface exposure calculation method proposed by [31] was used. As shown in Figure 4, the diameter of 8mm was marked at each pole (top vertex, bottom vertex, left, right, front, back) of the apple surface. The apple was rotated using a rotating mechanism to count the total number of poles $N_{t}$ and the number of poles that can be captured by the cameras $N_{c}$, and the surface exposure rate was calculated using Equation (1).
(1)$$exposure\;rate=\frac{N_{c}}{N_{t}},$$

A total of 50 randomly sampled normal and surface defective apples was calculated for surface exposure rate and repeated 5 times. Then, by calculating the average exposure rate, the surface exposure rate for each type could be calculated, as shown in Table 2. Spherical objects of imperfect shape were difficult to rotate perfectly with a rotating mechanism, resulting in a slightly lower surface exposure rate than other cases.

### 3.2. Data Acquisition

Using the system developed in Section 3.1, we captured a total of 12,000 images by taking six images per apple for 2000 apples (Fuji) harvested from an orchard in Jangseong-gun, South Korea. The acquired apple images were labeled into two classes, normal and surface defects apples, based on the criterion of whether or not there were surface defects, as shown in Figure 5. Apples with surface defects exhibited substantial deformations in their skin, such as irregular patterns, pest infestations, as well as morphological and physiological aberrations, attributable to multifaceted factors including climatic and environmental conditions. As shown in Figure 5b–e, the majority of these deformities were visually conspicuous and could be discerned with ease using unaided observation. Nonetheless, there were instances where apples may have sustained scratches on their surface during the harvesting process due to physical contact. Such scratches, as shown in Figure 5f, tended to be relatively small in size compared to other surface defects, and their features may be obscured by the reflection of light or insufficient illumination, making them a challenge to classify accurately. For this reason, apples with fine scratches were more likely to be misclassified, which could result in ambiguous labeling at the boundary between the two classes. To ensure accurate classification, we adopted a rigorous manual labeling process with repeated verification. Any images that were difficult to classify based on visual inspection alone were excluded from the surface defects apple class and labeled as normal apples to avoid misclassification. As a result, we constructed a dataset named Surface Defect Apple (SDA) by obtaining a total of 12,000 images and excluding 2000 ambiguous images, resulting in a dataset of 10,000 images. This dataset comprised 6201 normal-class images and 3799 surface defects-class images.

Meanwhile, due to the use of a 5 MP camera in the system, the image resolution of the dataset was 2440 × 2048, which was somewhat large for training a CNN classifier. As a common practice, downsampling is a straightforward method of resizing images to adjust their size when training with extremely large images. However, downsampling apple images captured with the background may result in the loss of critical features, such as fine scratches or defects, as they may become indiscernible or blurred. Therefore, downsampling in this manner could cause the images to become similar to normal apples, increasing the probability of misclassification during training and resulting in the learning of incorrect class information. Thus, additional image preprocessing is necessary to ensure uniformity and feature preservation across various types of apples in high-quality images.

### 3.3. Data Preprocessing

In CNN-based object classification, the presence of background noise can significantly impact the model’s generalization performance and lead to inconsistent classification results [32]. The acquired images of apples contained non-target background elements that could further deteriorate the model’s performance. Moreover, the original image size was 2440 × 2048 × 3, making it computationally expensive to use the embedded board for direct inference using a CNN classifier. However, forcing all images to be resized equally could lead to microscopic scratches being categorized into the same class as normal apples, which was a critical issue that must be handled with caution. Therefore, we propose an apple segmentation algorithm that extracts only the apple area.

Step 1. Pyramid-based apple position estimation: Given an input image I in RGB color of size $h_{old}\times w_{old}$, the pyramid downsampling method is applied to reduce the image size while maintaining the apple shape. This method enables fast and efficient apple detection, and the calculation is based on Equation (2) when the maximum size of the reduced image is assumed to be S.
(2)$$ratio=[{log}_{2}(\frac{max(h_{old}\times w_{old})}{S})],$$

The ratio value is the number of times the pyramid downsampling method is applied to I. We experimentally fit three times for pyramid downsampling. Thus, we have $I_{reduce}$ of reduced size $h_{new}\times w_{new}$, where $max\left(h_{old}\times w_{old}\right)=S$. However, in the first-step approach, it is difficult to obtain an apple with a perfect boundary because apple body is greatly affected by the segment when there is a region such as background noise or an apple stem. Therefore, step 2 is performed.

Step 2. Histogram-based complete apple segment: We apply a Gaussian blur filter with σ=2 to the image $I_{reduce}$ to remove noise for better apple region estimation. Additionally, $I_{reduce}$ uses a color-based approach to estimate the apple position. Thus, we assume that R, G, and B are matrices of red, green, and blue channels in $I_{reduce}$. Therefore, it is possible to check the intensity of the R, G, and B pixels at the horizontal coordinates, aligned with the center coordinate values of the apple, as shown in the sample apple image in Figure 6a, which is depicted as shown in Figure 6b. And for each color, the red line stands for R, the green line for G, and the blue line for B.

In this distribution, as shown in Equation (3), in the area of the apple, the red value is much higher than the green value, the difference is higher than 40, and the red value is higher than the blue value. Therefore, the region satisfying this is estimated to be the apple region.
(3)$$\left\{\begin{array}{c} R-G\geq 40\\ R-B>0 \end{array}\right.,$$

As shown in Figure 6, to remove the small blob created outside the apple area and to connect the area that is not completely closed, the apple boundary is smoothly filled using morphology with an ellipse kernel of size 9 × 9, and the apple boundary is selected the largest area among the filled areas. Then, if only the outer apple boundary is connected by applying the convex hull algorithm, only a perfect apple region is finally extracted. The finally extracted apples are as shown in Figure 7d.

Step 3. Crop the apple area: we estimate the smallest rectangle that encloses the apples, returning a 4-dimention of type (x,y,w,h), where (x,y) is the upper-left coordinate and (h,w) is the height and width of the rectangle. In general, the input image for training a CNN model is a square. We normalize this vector so that it has the smallest square that binds the complete segmented apples obtained in step 2. If h is higher than w, Equation (4) is followed. Otherwise, h is lower than w according to Equation (5). We cut the apple directly from the original I by adjusting this vector of apple area back to the new vector bbox. For the convenience of cropping the image, bbox is of type $\left(x_{1},y_{1},x_{2},y_{2}\right)$, and, in Equation (6), $\left(x_{1},y_{1}\right)$ is the upper-left coordinate and $\left(x_{2},y_{2}\right)$ is the lower-right coordinate.
(4)$$\left\{\begin{array}{c} x≔\left\lfloor x-\frac{h-w}{2}\right\rfloor \\ w≔h \end{array}\right.,$$
(5)$$\left\{\begin{array}{c} y≔\left\lfloor y-\frac{w-h}{2}\right\rfloor \\ h≔w \end{array}\right.,$$
(6)$$\left\{\begin{array}{c} \begin{array}{c} x_{1}≔\left\lfloor 2^{ratio}\cdot \left(x-\frac{h_{new}}{2}\right)+\frac{h_{old}}{2}\right\rfloor \\ y_{1}≔\left\lfloor 2^{ratio}\cdot \left(y-\frac{w_{new}}{2}\right)+\frac{w_{old}}{2}\right\rfloor \end{array}\\ \begin{array}{c} x_{2}≔\left\lfloor 2^{ratio}\cdot \left(x+h-\frac{h_{new}}{2}\right)+\frac{h_{old}}{2}\right\rfloor \\ y_{2}≔\left\lfloor 2^{ratio}\cdot \left(y+w-\frac{w_{new}}{2}\right)+\frac{w_{old}}{2}\right\rfloor \end{array} \end{array}\right.,$$

### 3.4. Knowledge-Distillation-Based Lightweight CNN Classifier

The relationship between the size of deep learning model parameters and their performance has been found to be directly proportional, indicating that an increase in the number of parameters generally leads to improved results. However, developing a lightweight model with an optimal balance between parameter size and performance, particularly in constrained hardware environments such as the system proposed in this paper, is a challenging task. Accordingly, various approaches have been proposed to address this issue. We utilize knowledge distillation (KD) [33] as a method for developing a lightweight CNN classifier that can efficiently operate within the limited resources of the target hardware system while maintaining high accuracy.

The training process for KD is similar to that of general deep learning training, with one distinction. In traditional deep neural network training, a softmax function is applied to generate a probability value for each class before the final layer. The softmax function is expressed as follows, assuming the logit is $z_{i}$, and the probability of each class is $q_{i}$.
(7)$$q_{i}=\frac{exp(z_{i})}{\sum_{i}exp(z_{i})},$$

Softmax is commonly used as an activation function in deep learning, including in knowledge distillation. However, the use of softmax in knowledge distillation can pose a problem due to the fact that it maps the largest input value close to 1 and the remaining inputs close to 0, which can cause even minor variations in input values to result in significant output changes. This can reduce the impact of smaller input information and compromise the regularization effect.
(8)$${softmax(q}_{i})=\frac{exp(z_{i}/\tau )}{\sum_{i}exp(z_{i}/\tau )},$$

To overcome this problem, a hyperparameter known as Temperature (τ) is introduced in knowledge distillation as shown in Equation (8). This parameter scales the softmax function, producing a softer probability distribution as its value increases. When τ=1, the distribution corresponds to the standard softmax function. Increasing τ leads to a softer distribution, where lower probabilities are given more weight, and high probabilities are given less emphasis.

Kullback–Leibler divergence (KL divergence) is a measure of how one probability distribution differs from another probability distribution. In KD, KL divergence is used to calculate the difference between the soft-target probability distribution obtained from the teacher model and the ground truth distribution. Thus, during the training process, the teacher model provides a soft-target probability distribution as a guide for the student model to learn from. This soft-target distribution is typically obtained by applying a softmax activation to the output of the teacher model. The student model then learns to approximate the soft-target distribution using its own output.

KL divergence is used to measure the difference between the soft-target distribution and the ground truth distribution. The ground truth distribution represents the true probability distribution of the target labels. The KL divergence measures the amount of information lost when the soft-target distribution is used to approximate the ground truth distribution.
(9)$$L_{soft}=\sum_{x_{i}\in X}KL(softmax\left(\frac{f_{t}(x_{i})}{\tau }\right),softmax\left(\frac{f_{s}(x_{i})}{\tau }\right)),$$
(10)$$L_{task}=CrossEntropy\left(softmax\left(f_{t}\left(x_{i}\right)\right),y_{truth}\right),$$
(11)$$Student\;L_{total}=L_{task}+\lambda \cdot L_{soft},$$

By minimizing the KL divergence between the soft-target distribution and the ground truth distribution, the student model is encouraged to learn from the teacher model’s knowledge as shown in Equations (9)–(11). Here, $f_{t}$($x_{i}$) is the logit value of the teacher model, and $f_{s}(x_{i}$) is the logit value of the student model in Equation (9). In this way, KD can help the student model achieve better performance than it would have otherwise by leveraging the knowledge contained in the teacher model.

Meanwhile, the teacher refers to the large deep learning model, while the student refers to a smaller model that aims to learn from the teacher’s knowledge. Specifically, the teacher model transfers its learned weights and biases to the student model to assist its training. This knowledge transfer is then utilized by the student model to learn from its own set of data. From this perspective, it can be argued that the performance of knowledge distillation in image recognition tasks is heavily dependent on the performance of the teacher model. Considering this, we have selected the well-trained deep neural networks EfficientNet [34] and RegNet [35], which have demonstrated impressive performance on datasets such as CIFAR-10, CIFAR-100, SVHN, and ImageNet, to serve as our teacher models. In this study, we fine-tune the pre-trained teacher model on the ImageNet dataset to train SDA dataset.

For the teacher model, we chose both large and medium-sized models (EfficientNet-B4 and B7 and RegNetY-8.0GF and 3.2GF) from EfficientNet and RegNet. Additionally, we then trained the student model ResNet18 [36] on the same dataset using the KD framework, as depicted in Figure 8. The student model, which was trained using the KD process, fulfills the role of a CNN classifier in our system.

According to TensorFlow Hub, ResNet50 is the most popular among large-capacity models that achieve high performance, as it has demonstrated stable performance in various research and development projects [37]. Therefore, we considered using ResNet50 in our system, but it is already a large model to be used on our embedded board. As a result, we used the shallowest model in the ResNet family ResNet18 and designed a smaller-sized model than the original ResNet18 for use in the KD process. The design of this model is as follows:

The student model ResNet18 is a deep neural network consisting of 18 layers and 5 blocks (A, B, C, D, E), along with fully connected layers. Each block, except for blocks A and B, is composed of a Convolution Block and an Identity Block. We adopt the ResNet18 architecture but reduce all channel sizes to half for a lightweight model, as illustrated in Figure 9. As a result, the output feature map from Block A is 56 × 56 × 32. The feature map sequentially passes through each layer, and adaptive average pooling is applied to obtain a 7 × 7 × 256 feature map, which is fully connected to two neurons. Finally, the softmax function is utilized to determine the probability of belonging to each of the two classes based on the output values of the two neurons.

## 4. Experiment Results

### 4.1. Experimental Setup

In this study, all experiments were conducted on a 64-bit Ubuntu 22.04.01 LTS operating system with 32 GB of RAM, an Intel^®^ Core™ i9-10900X @ 3.70 GHz CPU, and an NVIDIA Quadro RTX8000 with 48 GB of memory. All code was implemented using the PyTorch framework, an open-source deep neural network library written in the Python language.

To deploy a CNN classifier on an embedded board, the model must be optimized for the GPU environment. To accomplish this, we utilized TensorRT, which can import and deploy models from any deep learning framework. TensorRT-based applications can perform up to 40 times faster than CPU-only platforms [38]. To enable real-time inference with the pre-trained model on the embedded board, we converted the model to the Open Neural Network Exchange (ONNX) format [39]. The Lightweight CNN Classifier, the student model trained through KD, was converted to the ONNX format and deployed into the system’s embedded board.

To quantitatively evaluate the performance of the proposed system, we verified the accuracy and speed of a CNN classifier trained on the SDA dataset collected by the system. The dataset consisted of 10,000 images and was smaller in quantity compared to typical large-scale datasets. Furthermore, it was imbalanced, with 6201 normal images and 3799 defective apple images. To improve the reliability of the model trained on a small and imbalanced dataset, we employed Stratified K-Fold cross-validation. Specifically, we divided the dataset into a training set consisting of 7000 images and a testing set consisting of 3000 images, with equal numbers of normal and defective images in the testing set (1500 each). The remaining 7000 images in the training set had a class distribution of 67.15% (4701 images) normal and 32.85% (2299 images) defective. To perform Stratified K-Fold cross-validation, we needed to ensure that the class distribution was maintained in each fold, with nearly equal numbers of normal and defective images. Therefore, we partitioned the training set into 5 folds, as shown in Table 3, with a ratio of 4:1 for the training and validation data in each fold.

The original size of the dataset’s images was 2448 × 2048, and we applied the proposed apple segmentation preprocessing algorithm to crop only the apple region. We then downsampled the cropped images to 576 × 576 to train the CNN classifier. Additionally, generally, data augmentation techniques such as image transformation and noise addition are needed to improve the generalization performance of deep learning models. However, these techniques were unsuitable for the dataset used in this experiment. The main reason is that excessive use of transform and noise can induce variations in apple color and shape, leading to incorrect biases in the model. Therefore, we did not use any data augmentation techniques in this experiment.

The first layer weights of the teacher model were initialized with pre-trained ImageNet weights. Additionally, we determined the optimal hyperparameters, including a learning rate of 0.001, a batch size of 32, and 100 epochs, through various experiments. We used cross-entropy as the loss function for the teacher model and Equation (11) for the student model, with λ set to 0.7 and τ set to 3. The LAMB optimizer [40] was used for training. For each student model, we conducted KD experiments with two teacher models. In the first experiment, we trained the SDA dataset using each pre-trained teacher model and selected the model with the highest performance among the five-fold cross-validation models. In the second experiment, we demonstrated the superior performance of the student model trained through KD on the same dataset using the previously trained teacher model.

The performance of the student model (lightweight CNN classifier) obtained through KD was evaluated using Accuracy, Precision, Recall, and F1 for each apple class i, is as follows in Equation (12):(12)$$Accuracy=\frac{{TP}_{i}+{TN}_{i}}{{TP}_{i}+{TN}_{i}+{FP}_{i}+{FN}_{i}},\;Precision=\frac{{TP}_{i}}{{TP}_{i}+{FP}_{i}},i=1,2;\;Recall=\frac{{TP}_{i}}{{TP}_{i}+{FN}_{i}},\;F1=2\frac{Precision\cdot Recall}{Precision+Recall}$$
where true positive (${TP}_{i}$) is the number of true positive classifications for class i, true negative (${TN}_{i}$) is the number of true negative classifications for class i, false positive (${FP}_{i}$) is the number of false positive classifications for class i, and false negative (${FN}_{i}$) is the number of false negative classifications for class i.

### 4.2. Results and Discussion

In this section, we examined three key performance metrics: the performance of the teacher, KD-based student models that will be used as CNN classifier, and the performance of the system when deploying the acquired student model. To accomplish this, we first conducted training and evaluation of the teacher model for the SDA dataset-based KD. Then, we selected one teacher model and used it to train and evaluate the student model through KD. Additionally, we integrated the acquired lightweight CNN classifier into the system and evaluated its performance in terms of the time and accuracy required for sorting apples.

#### 4.2.1. Performance of Teacher Model

The performance of the teacher model was evaluated, and the RegNetY-8.0GF and RegNetY-32GF models achieved the highest accuracy of 97.07% and 97.57%, respectively, with F1s of 95.91 and 97.23, as shown in Table 4.

The EfficientNet-B4 and EfficientNet-B7 model achieved an accuracy of 95.57% and 96.86%, respectively, with F1-scores of 94.98 and 96.45. All models used proposed apple segmentation preprocessing algorithm to improve their performance. Among all the models, RegNetY-32GF achieved the highest accuracy. Table 5 shows that the RegNetY-32GF model has the largest number of parameters at 141.34 M.

#### 4.2.2. Performance of Student Model

In this section, we demonstrate the effectiveness of KD in reducing the parameter size of the student model while maintaining high accuracy on the SDA dataset. To demonstrate this, we used the teacher models, trained in Table 4, to distill knowledge to the student models and evaluated their performance. We also trained models without KD to examine the effectiveness of KD on the SDA dataset. For clarity, models trained with KD were labeled as “w/ KD”, while those trained without KD were labeled as “w/o KD”. The teacher models were named RegNetY-8.0GF (R8), RegNetY-32GF (R32), EfficientNet-B4 (EB4), and EfficientNet-B6 (EB6), and ResNet18 was used as the student model.

Table 6 presents the results of the experiment, showing that the RegNetY-8.0GF teacher model achieved the highest accuracy of 94.79% among the ResNet18-based student models trained with KD. In comparison, the original ResNet18 model achieved an accuracy of 80%, which was lower than the models trained with KD (−14.79%). While we expected that the student model using the largest model, RegNetY-32GF (94.71%), would achieve the highest accuracy, the result was different. The loss and accuracy for all student models’ experiments can be seen for each in Figure 10. KD-R32 (orange) does not show a visible curve on the graph, but overlaps closely between KD-EB4 (green) and KD-EB7 (red), which appear very similar to each other.

Therefore, using the student model (ResNet18 + w/KD-R8) obtained in the end, the confusion matrix for the test set in Figure 11 is as follows. The Recall value of the model is marginally higher than the Precision value, indicating the model prioritized not to miss any surface defects samples but misclassified some normal samples as surface defects.

As a result, the CNN classifier (student) obtained through KD not only guarantees higher performance than when not using KD but also reduces the number of model parameters by approximately 3.9 times, as shown in Table 7.

The results of this simple experiment demonstrate the effectiveness of KD in significantly improving the performance of ResNet18-based student models on the SDA dataset. The comparison between models trained with and without KD clearly shows the effectiveness of knowledge transfer from the teacher models to the student models. Using high-capacity models such as RegNetY-32GF(R32) might seem to be an evident choice for achieving superior performance in KD. However, experimental results have shown that RegNetY-8.0GF can actually outperform such models, indicating that a high-capacity model is not always necessary in KD [41].

#### 4.2.3. Performance of System

The CNN classifier (ResNet18 + W/KD-R8) obtained through KD was deployed in the system, and its classification performance was tested by randomly collecting 300 samples in two scenarios.

For the first scenario, the performance evaluation was focused on the inference speed and accuracy of the CNN classifier deployed in the system. To conduct this evaluation, 300 random apple samples were selected for testing. The selected sample was utilized to evaluate the efficiency of the inference process of a CNN classifier in terms of classification accuracy and speed. By carefully examining and presenting the performance results obtained, as shown in Table 8, valuable insights were provided into the practicality of the system in real-world environments. This performance evaluation plays a crucial role in evaluating the suitability and efficiency of the CNN classifier in actual applications where optimal accuracy and speed are crucial for success.

To evaluate the entire system process, we controlled the stepping motor that rotated the hardware mechanism responsible for holding the apple, which affected the quality of image acquisition. We tested the performance of the CNN classifier according to the optimal stepping motor speed and evaluated to find the best speed. As shown in Table 9, we limited the stepping motor to a range of 0.5 to 0.7 s, as apples became unstable and fell off the holder if rotated faster than 0.5 s. Additionally, the experimental results showed that no significant difference was observed beyond 0.7 s, with similar results obtained under identical conditions. When the speed of the stepping motor was too fast, it took longer for the apple to be stable on the holder, resulting in acquiring blurred images that led to a significant decrease in accuracy. Although the processing speed per apple could be faster when the speed of the stepping motor was faster, the accuracy was significantly reduced. Therefore, controlling the stepping motor speed with an interval of 0.7 s allowed each of the three CNN classifiers in embedded board to perform two-time classifications, including segmentation preprocessing speed, requiring 0.14 s, resulting in an overall accuracy of 93.83%. Considering the waiting time of 2 s for the apple to settle on the holder, it took an average of 2.84 s to sort one apple.

The apple sorting systems in Table 1 are diverse, but the system that employs a CNN classifier is the same as that presented in Table 10. When compared to previous studies that solely employed a CNN classifier, Jijesh, J.J. et al. demonstrated higher performance (+2.83%) than our system. However, as the system only used one camera and captured images of only one side of the apple without any rotation, it could be considered highly inaccurate. Thus, the major advantage of our system is the ability to scan the surface of the apple through regular balanced rotation and classify it with high accuracy. Additionally, the system is optimized for apple sorting, as explained in Section 3.2, and the camera structure can be flexibly changed, enabling the system to be applied to sorting other fruits as well.

Fuji apples typically weigh around 150 g to 200 g, and one metric ton of Fuji apples contains approximately 5000 to 6660 apples. Assuming an average of 5830 Fuji apples per metric ton, sorting one metric ton of apples using our system in the orchard takes approximately 4 h and 35 min. The proposed system is based on a single conveyor belt, and, if three conveyor rails with the same system are built, sorting one metric ton of apples would take only about 1 h and 30 min. Using the proposed system, orchards with high apple yields of about 100 metric tons can benefit from its high accuracy and speed, with a maximum processing time of approximately 6 days. While this system is optimized for apple sorting, as described in Section 3.2, its camera structure can be flexibly modified, making it applicable for sorting other fruits as well.

## 5. Conclusions

In this study, we introduced a novel apple defect detection system that incorporated a rotation mechanism and a multi-camera setup to effectively address the limitations of traditional single-camera systems. The proposed system ensured uniform and accurate imaging of the entire apple surface by rotating individual apples while simultaneously capturing their images with multiple cameras. This approach allowed for a comprehensive and efficient analysis of the apple surface compared to the conventional single-camera techniques. To further enhance the system’s performance, we employed a lightweight CNN classifier, which was trained using knowledge distillation techniques. This approach enabled the classifier to maintain excellent performance, even on low-spec embedded boards, making it an ideal solution for real-world applications. We also developed a segmentation algorithm that effectively trained the model by focusing solely on the apple regions, ensuring a more accurate defect detection process. With the current processing sequence, the system demonstrates a processing speed of 2.84 s per apple and an accuracy rate of 93.83%. However, by modifying the processing sequence to immediately classify an apple as defective upon detecting a defect, the system’s processing speed can be further improved, enhancing its overall efficiency. In the end, this system can classify individual apples with greater precision and accuracy than previously proposed systems.

One significant advantage of the proposed system, which has not been discussed in this paper, is its flexibility and potential applicability to various fruits. In future research, we plan to explore image collection and classification processing for a wide range of fruits using the proposed system. Additionally, we will investigate apple grading classifications that require more precise observation beyond simple defect detection, further expanding the system’s capabilities and applications in the fruit industry.

## Figures and Tables

**Figure 1 sensors-23-03968-f001:**
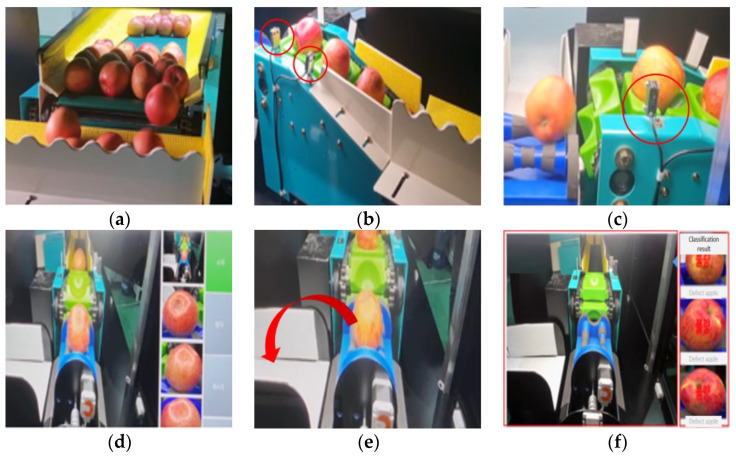
The process of the apple sorting system: (**a**) apple supply conveyor; (**b**) transfer conveyor belt; (**c**) object detection sensor; (**d**) processing of CNN classifier; (**e**) stepping motor; (**f**) monitoring of classifier result.

**Figure 2 sensors-23-03968-f002:**
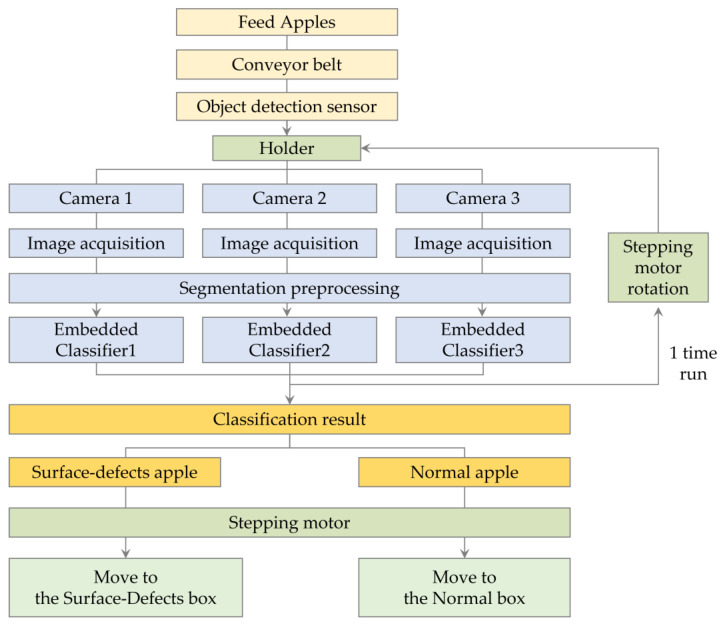
The workflow of the defective apple sorting system.

**Figure 3 sensors-23-03968-f003:**
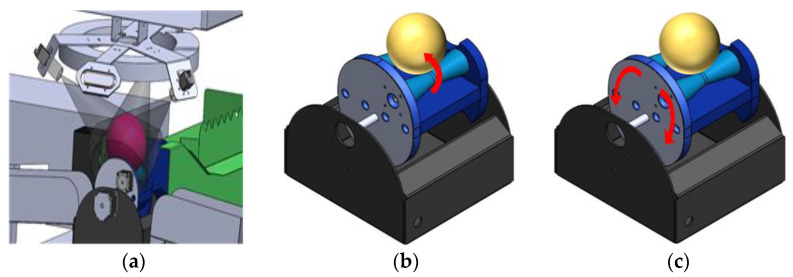
Camera structure and rotation mechanism: (**a**) Camera structure; (**b**) Stepping motor for apple rotation; (**c**) Apple sorting based on the results of a CNN classifier.

**Figure 4 sensors-23-03968-f004:**
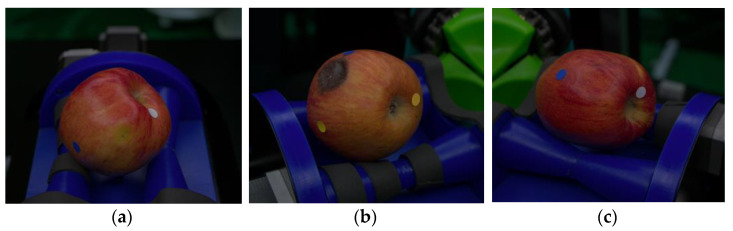
Example of sampled apples for surface exposure calculation: (**a**) Normal; (**b**,**c**) Surface-defects apple.

**Figure 5 sensors-23-03968-f005:**
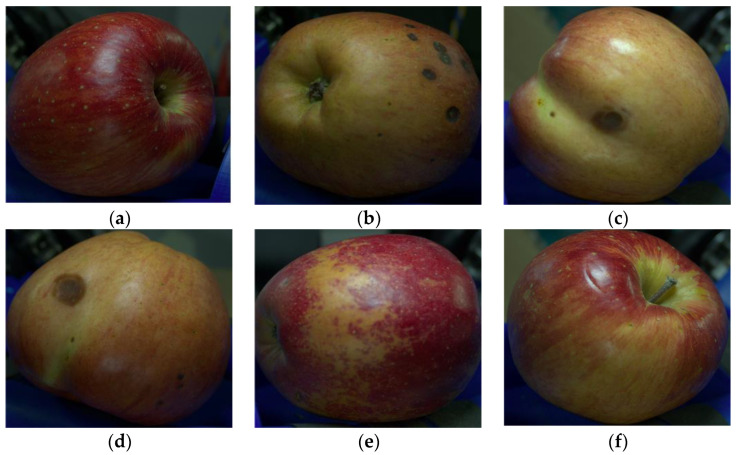
Normal and surface defects apple cases: (**a**) Normal; (**b**) Physiological disorder; (**c**) Malformation; (**d**) Pest; (**e**) Russeting; (**f**) Scratch.

**Figure 6 sensors-23-03968-f006:**
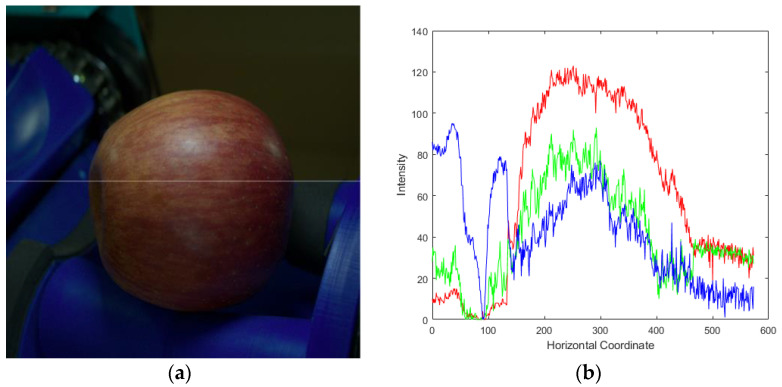
Example of analysis of the RGB histogram of apples: (**a**) sample image; (**b**) RGB histogram.

**Figure 7 sensors-23-03968-f007:**
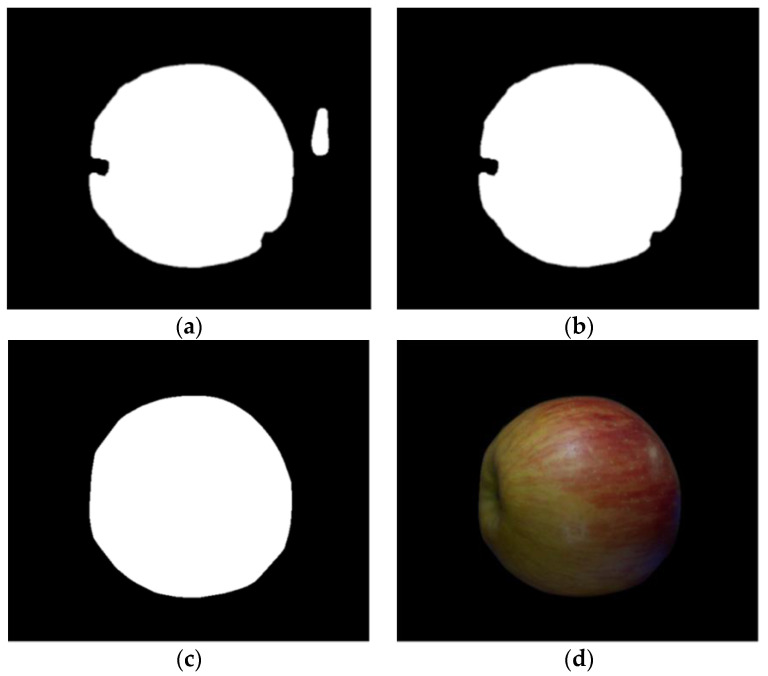
Extracting apple area from a sample image: (**a**) Morphology result; (**b**) Large area selection; (**c**) Convex hull algorithm; (**d**) Apple area.

**Figure 8 sensors-23-03968-f008:**
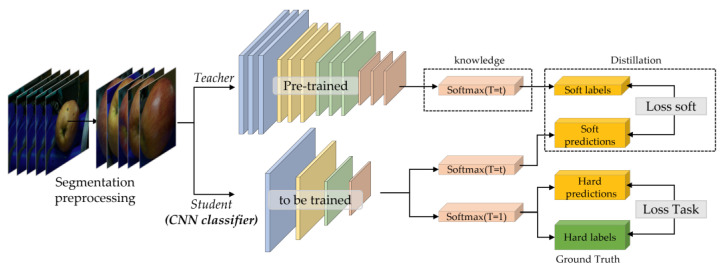
KD framework for CNN classifier.

**Figure 9 sensors-23-03968-f009:**
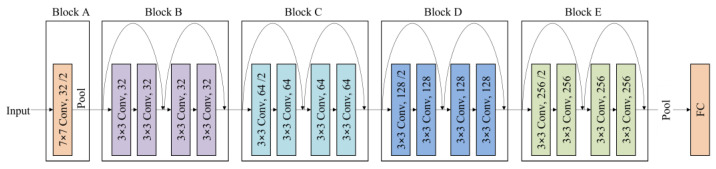
ResNet18 architecture with half the parameters for lightweighting.

**Figure 10 sensors-23-03968-f010:**
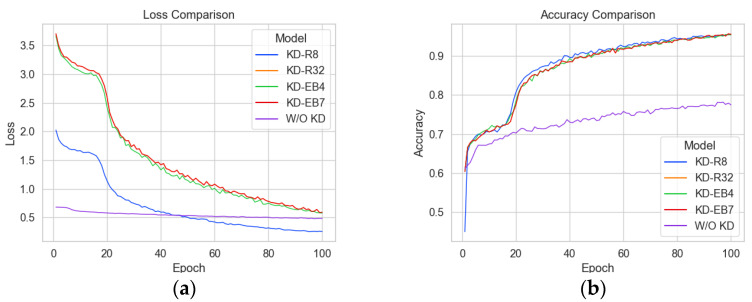
Loss and Accuracy for all student models: (**a**) Loss; (**b**) Accuracy.

**Figure 11 sensors-23-03968-f011:**
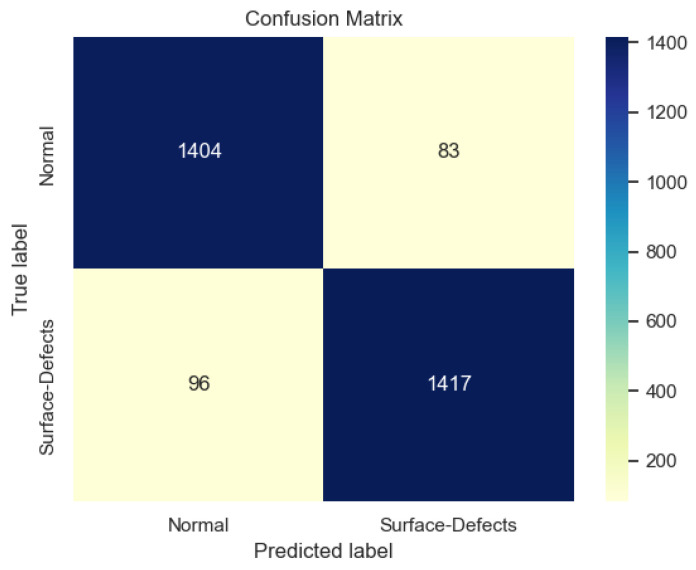
Confusion Matrix for Test Dataset.

**Table 1 sensors-23-03968-t001:** Apple sorting system utilizing mechatronic components (2010–2023).

System Type	Authors	Year	Rotation Type	Conveying Type	Number of Camera	Classifier Type	Accuracy
M-R	Xiao-bo, Zou et al. [20]	2010	Free rotation	conveyor	3	image processing techniques	89%
M-R	Sofu, M.M. et al. [21]	2016	Free rotation	roller conveyor	2	image processing techniques	96%
M-N/R	Fan, Shuxiang et al. [22]	2020	N/A	roller conveyor	2	CNN Classifier	92%
S-N/R	Jijesh, J.J. et al. [23]	2022	N/A	conveyor	1	CNN classifier	96.66%
M-R	Hu, Guangrui et al. [24]	2021	Free rotation	conveyor	3	SVM + SSD	94.12%
M-R	Fan, Shuxiang et al. [25]	2022	Free rotation	conveyor (black rubber rollers)	4 (two CCD, two NIR)	YOLO V4	93.9%
S-R	Kuo, Hang Hong et al. [26]	2022	N/A	conveyor	1	SSD	80%
S-R	Liang, Xiaoting et al. [27]	2022	Free rotation	fruit tray	1	Segmentation + YOLO V4	92.42%
S-R	Endo, Minori et al. [28]	2022	N/A	conveyor	1	CNN Classifier	84.1%
S-N/R	Xu, Bo et al. [29]	2023	N/A	conveyor (black rubber rollers)	1	Attention module + YOLOv5	93%

**Table 2 sensors-23-03968-t002:** Quantitative surface exposure rate for each type of apple.

Type	Normal	Physiological Disorder	Malformation	Pest	Russeting	Scratch
ER *, %	100%	100%	96%	100%	100%	100%

* ER—exposure rate.

**Table 3 sensors-23-03968-t003:** Data splitting according to five-fold Stratified K-Fold cross-validation on imbalanced dataset.

Fold	Train-Set (Normal/Defective)	Validation-Set (Normal/Defective)
1	5600 (3760/1840)	1400 (940/460)
2	5600 (3760/1840)	1400 (940/460)
3	5600 (3760/1840)	1400 (940/460)
4	5600 (3760/1840)	1400 (940/460)
5	5600 (3760/1840)	1400 (940/460)

**Table 4 sensors-23-03968-t004:** Performance of the teacher models.

Teacher Model	Accuracy	Precision	Recall	F1
RegNetY-8.0GF	97.07	96.56	96.82	95.91
RegNetY-32GF	97.57	97.45	97.02	97.23
EfficientNet-B4	95.57	94.93	95.03	94.98
EfficientNet-B7	96.86	96.30	96.60	96.45

**Table 5 sensors-23-03968-t005:** Number of parameters and computational complexity of the teacher models.

Teacher Model	Pram *	CC *
RegNetY-8.0GF	37.17	52.72
RegNetY-32GF	141.34	213.52
EfficientNet-B4	17.55	10.14
EfficientNet-B7	40.74	22.56

* Param—parameter size, million; CC—computational complexity, GMACs.

**Table 6 sensors-23-03968-t006:** Performance of the student models based on KD.

Student Model	Accuracy	Precision	Recall	F1
ResNet18 + w/KD-R8	94.79	94.48	93.61	94.03
ResNet18 + w/KD-R32	94.71	94.88	93.05	93.88
ResNet18 + w/KD-EB4	93.79	94.46	91.42	92.71
ResNet18 + w/KD-EB7	93.36	93.86	90.99	92.22
ResNet18 + w/o KD	80.00	79.26	73.46	75.12

**Table 7 sensors-23-03968-t007:** Comparison model parameters and computational complexity.

Student Model	Param *	CC *
ResNet18 + w/KD-R8	2.8 M	6.44
ResNet18 + w/o KD	11.18 M	24.10

* Param—parameter size, million; CC—computational complexity, GMACs.

**Table 8 sensors-23-03968-t008:** Scenario 1: Inference speed and accuracy of a CNN classifier on an embedded board.

Student Model	Inference Speed	Accuracy
ResNet18 + w/KD-R8	0.069	93.21
ResNet18 + w/o KD	0.168	79.29

**Table 9 sensors-23-03968-t009:** Scenario 2: Performance evaluation of the overall system according to the control of stepping motor.

Stepping Motor Speed (m/s)	Accuracy	Total Processing Time * (m/s)
0.5	86.52	2.44
0.6	88.42	2.64
0.7	93.83	2.84

* One apple sorting time through the system.

**Table 10 sensors-23-03968-t010:** Performance comparison with systems based on CNN classifier.

System Type	Authors	Rotation Type	Conveying Type	Number of Camera	Classifier Type	Accuracy
S-N/R	Jijesh, J.J. et al. [23]	N/A	conveyor	1	CNN classifier	96.66%
M-N/R	Fan, Shuxiang et al. [22]	N/A	roller conveyor	2	CNN Classifier	92%
S-R	Endo, Minori et al. [28]	N/A	conveyor	1	CNN Classifier	84.1%
M-R	Proposed system	Regular rotation	conveyor	3	CNN Classifier + Knowledge distillation	93.83%

## Data Availability

The Surfaces defects of apples is available online at https://github.com/dev-johnlee/SD-Apples-with-KD/tree/main/dataset (accessed on 11 April 2022).

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
