# Peer review of "Multi-Camera-Based Sorting System for Surface Defects of Apples"

_sensors, 2023, doi:10.3390/s23083968_

Round 1

Reviewer 1 Report

1- authors must provide a literature survey.

2- authors must provide a table of the recent studies.

3- authors must provide a comparison between their model and the recent studies.

4- authors must explain the dataset, the number of each class, and the split of the dataset.

Reviewer 2 Report

The paper presents an interesting subject, but the following aspects must be detailed more clearly:

-not clearly if the novelty of the paper consists in the device development or the proposed method for defect detection (or both)

-there is no state of the art section for describing other existing aspects that are correlated with this subject (including also the existing results)

-how was chosen the architecture for CNN: what are the main aspects that were considered: existing results, number of parameters, etc

-what types of defects are detected? a confusion matrix for these defects must be added in order to show what are the defects that are better detected.

Reviewer 3 Report

1. Figure 1 did not clearly display the sorting system.

2. The sorting process was not well described, and Figure 2 did not accurately depict the sorting process.

3. How were the six images captured? The process of shooting should be clearly described.

4. Figure 5(b) did not have x and y coordinates and the lines were unclear.

5. There were no CNN classifier results shown, and the changes in loss rate and accuracy during the classifier iteration process were not displayed.

6. The manuscript had too little data to support the conclusions.

Round 2

Reviewer 2 Report

Part of my comments were addressed but section related work must contain also obtained results in order to have an accurate comparison with results obtained by the proposed method.

Reviewer 3 Report

The manuscript proposed a multi-camera-based sorting system for surface-defects apple. The system provided an efficient and 27 precise solution for detecting defects on the entire surface of apples, thereby improving the sorting 28 process with high reliability. The research had some scientific significance and value. The manuscript can be taken into consideration for publishing in “Senonrs Journal” only if the following comments are addressed.

1.      The abstract should be rewritten. As we know, most or all of the Abstract should be written in the past tense, because it refers to work done.

2.      The comma at the end of keywords should be deleted.

3.      The label of references in the body of the manuscript is wrong, such as the label of the sentence -“To address this limitation, systems have been proposed [6,7,8,9]……” is wrong. The correct label should be “To address this limitation, systems have been proposed [6-9]……”.

4.      The format of some references was wrong and should be checked and corrected thoroughly, such as the symbols of “pp” in the reference should be deleted.

5.       The authors are suggested to completely revise the text for grammatical and writing errors.
